# CO<sub>2</sub> deviation in a cylinder due to consumption of a

# 2 standard gas mixture

- Nobuyuki Aoki<sup>1,3,2</sup> and Shigeyuki Ishidoya<sup>2,4,1</sup>
- <sup>1</sup>Integrated Research Center for Nature Positive Technology, National Institute of Advanced Industrial
- Science and Technology, Tsukuba, Ibaraki 305-8567, Japan
- <sup>2</sup>Global Zero Emission Research Center, National Institute of Advanced Industrial Science and Technology
- (GZR/AIST), Tsukuba 305-8569, Japan
- <sup>3</sup>National Metrology Institute of Japan, National Institute of Advanced Industrial Science and Technology
- (NMIJ/AIST), Tsukuba 305-8563, Japan
- <sup>4</sup>Research Institute for Environmental Management Technology, National Institute of Advanced Industrial

Abstract: The CO2 molar fraction in standard gas mixtures is known to deviate as a result of

Science and Technolog (EMRI/AIST), Tsukuba 305-8569, Japan

12

13

1

Correspondence to: Nobuyuki Aoki (aoki-nobu@aist.go.jp) Tel: +81-80-2232-2292

14

15

adsorption/desorption to/from the inner surface of a high-pressure cylinder and thermal diffusion 16 17 fractionation caused by the temperature distribution in the cylinder. This deviation reduces the consistency 18 of atmospheric CO<sub>2</sub> observations, because the standard gas mixtures are used to calibrate all measurement 19 systems for precise CO2 observations. To maintain the consistency of CO2 values over the long term, a 20 quantitative understanding of the deviations in the CO<sub>2</sub> molar fraction in a standard gas mixture is needed. 21 Thus far, this understanding has not been achieved sufficiently well, because the contribution of thermal 22 diffusion fractionation is less well understood than that of adsorption/desorption. In this study, offsets of  $0.013 \pm 0.015 \,\mu \text{mol mol}^{-1}$  and  $-0.014 \pm 0.011 \,\mu \text{mol mol}^{-1}$  were observed in the outflowing gas from 23 24 horizontally and vertically positioned cylinders, respectively, at a flow rate of 0.080 L min<sup>-1</sup>. These offsets 25 are attributed to thermal diffusion effects, which diluted and enriched the CO<sub>2</sub> molar fraction by -0.045 μmol mol<sup>-1</sup> (horizontal cylinder) and 0.048 μmol mol<sup>-1</sup> (vertical cylinder) as the relative pressure dropped 26 27 to 0.03. In the experiments at same flow rate, the adsorption/desorption effect enriched the CO<sub>2</sub> molar

- fraction by 0.06 μmol mol<sup>-1</sup> (horizontal cylinder) and 0.10 μmol mol<sup>-1</sup> (vertical cylinder). Therefore,
- attention should be paid to both thermal diffusion fractionation and adsorption/desorption effects for precise
- calibration of long-term observations of CO<sub>2</sub> molar fractions, although past studies have ignored the
- contribution of thermal diffusion fractionation at the low flow rates (<0.3 L min<sup>-1</sup>) examined in this study.
- Furthermore, the deviation of the CO<sub>2</sub> molar fraction depends only on the pressure relative to the initial
- pressure of the cylinder. This result suggests that the recommendation by the World Meteorological
- Organization (WMO) to replace the standard gas mixture once the cylinder pressure drops to 2 MPa needs
- to be revised.
- **Keywords:** standard gas mixture, atmospheric CO<sub>2</sub>, adsorption/desorption, thermal diffusion fractionation

Carbon dioxide (CO<sub>2</sub>) is an important greenhouse gas that contributes markedly to the radiative forcing of

# 10 1 Introduction

11

24

12 the atmosphere. Systematic observations of atmospheric CO2 have been conducted by numerous 13 laboratories around the world to better understand its sources and sinks. By determining the CO<sub>2</sub> molar 14 fraction in the atmosphere based on a scale established on the basis of primary standard gas mixtures in 15 high-pressure aluminum cylinders, the laboratories ensure consistency of the observed values over the long 16 term. Because deviations of the CO<sub>2</sub> molar fractions in the cylinders lead to over- or underestimation of the 17 measured CO<sub>2</sub> molar fraction and reduce the comparability of worldwide CO<sub>2</sub> observations, deviations of 18 the CO<sub>2</sub> molar fractions in the cylinders should be a focus of attention. 19 Langenfelds et al. (2005) reported that the air composition of a standard gas mixture in a high-pressure 20 cylinder could be modified by diffusive and surface processes. Subsequently, Leuenberger et al. (2015) and 21 Schibig et al. (2018) conducted "decanting experiments", in which a CO<sub>2</sub>-in-air mixture leaving a cylinder 22 was measured continuously, and found that the deviation of the CO<sub>2</sub> molar fraction in the cylinder could be 23 explained by adsorption/desorption phenomena to/from the cylinder inner surface. In the studies of

Leuenberger et al. (2015) and Schibig et al. (2018), the amounts of CO<sub>2</sub> adsorbed on the inner surface of

1 the cylinder, expressed as a fraction of the total gas in the cylinder, were estimated to be 0.028 µmol mol<sup>-1</sup> 2 and  $0.0165 \pm 0.0016$  µmol mol<sup>-1</sup>, respectively, in decanting experiments using 29.5 L aluminum cylinders. 3 Aoki et al. (2022) estimated the adsorbed  $CO_2$  molar fraction to be  $0.027 \pm 0.004 \,\mu\text{mol} \,\text{mol}^{-1}$  using 10 L 4 aluminum cylinders. Moreover, Schibig et al. (2018) reported that other effects such as thermal diffusion 5 fractionation became more pronounced than adsorption/desorption effects when the flow rate of the 6 outflowing gas from the cylinder was increased. Aoki et al. (2022) also suggested that thermal diffusion 7 fractionation was the main contributor to the "other effects" in their mother-daughter transfer experiments. 8 Aoki et al. (2022) and Schibig et al. (2018) pointed out that thermal diffusion fractionation depended on 9 the position of the cylinder: CO<sub>2</sub> molar fractions were enriched in vertically positioned cylinders but diluted 10 in horizontally positioned cylinders. Thermal diffusion fractionation is driven by the difference in the 11 diffusion velocity between CO2 and air caused by the temperature gradient in the cylinder, with heavier 12 molecules preferentially accumulating in colder regions. Therefore, these results suggest that colder air 13 leaves from horizontally positioned cylinders and warmer air leaves from vertically positioned cylinders. 14 The same series of primary standard gas mixtures should be used for as long a time as possible to maintain 15 consistency of the CO2 molar fractions. However, it is not possible to use standard gas mixtures down to 16 lower pressure because the CO<sub>2</sub> molar fraction in the cylinder deviates as the pressure drops as a result of 17 adsorption/desorption and thermal diffusion effects. Therefore, the World Meteorological Organization 18 (WMO) recommends that the standard gas mixtures should be replaced once the cylinder pressure has 19 decreased to 2 MPa. Leuenberger et al. (2015) and Schibig et al. (2018) recommended that the usage of 20 standard gas mixtures in aluminum cylinders should be restricted to pressures above 3 MPa to remain within the WMO's compatibility goal of  $0.1 \mu \text{mol mol}^{-1}$  for the northern hemisphere and  $0.05 \mu \text{mol mol}^{-1}$  for the 21 22 southern hemisphere. If the deviation of the CO<sub>2</sub> molar fraction could be corrected, standard gas mixtures 23 could be used down to lower pressure than the recommended value. However, currently it is difficult to 24 apply this correction because the magnitude of thermal diffusion fractionation has not been sufficiently 25 evaluated, in contrast to the considerable work on adsorption/desorption in previous studies.

- In this study, we quantitatively estimated the deviation of the CO<sub>2</sub> molar fraction in 10 L aluminum
- cylinders as the pressure dropped. First, CO<sub>2</sub> deviations were evaluated by means of decanting experiments
- with different flow rates of the outflowing gas. Second, the fractionation factors of CO<sub>2</sub> resulting from
- thermal diffusion fractionation were determined by subtracting the adsorption/desorption effect from the
- deviation in the CO<sub>2</sub> molar fraction measured in the decanting experiment. Last, the actual offsets of the
- CO<sub>2</sub> values caused by thermal diffusion effect were compared with the offset values calculated based on
- the fractionation factors. In addition, we discussed how the standard gas mixture in the cylinder should be
- operated based on the results obtained in this study.

#### 2 Methods

17

# 2.1 Experiment

#### 2.1.1 Sample gas mixtures

- 12 CO<sub>2</sub>-in-air mixtures were used as a sample gas to measure the deviations of CO<sub>2</sub> molar fractions. The
- mixtures were prepared by mixing pure CO<sub>2</sub> (>99.995 %, Nippon Ekitan Corp., Japan) with purified air
- 14 (G1-grade, <0.1 μmol mol<sup>-1</sup> for CO, CO<sub>2</sub>, THC, <0.01 μmol mol<sup>-1</sup> for NO<sub>x</sub>, SO<sub>2</sub>, < -80 °C for H<sub>2</sub>O, Japan
- 15 Fine Products, Japan) into a 10 L aluminum cylinder (Luxfer Gas Cylinders, UK;). The CO<sub>2</sub> molar fractions
- in the CO<sub>2</sub>-in-air mixtures were adjusted to an atmospheric level.

### 2.1.2 Decanting experiment

- 18 The CO<sub>2</sub>-in-air mixtures in 10 L aluminum cylinders positioned horizontally and vertically were decanted
- 19 from 10.0 MPa to 0.3 MPa at outflowing gas rates of 0.080 L min<sup>-1</sup>, 0.15 L min<sup>-1</sup>, 0.30 L min<sup>-1</sup>, 1.2 L
- 20 min<sup>-1</sup>, and 6.0 L min<sup>-1</sup>. A schematic diagram of the decanting experiment is shown in Fig. 1. The mixture
- 21 leaving the cylinder via a single-stage regulator (Torr 1300, NISSAN TANAKA Co., Japan) was divided
- 22 into two by means of T-pieces. The branched flows were controlled using two mass flow controllers, one
- 23 of which (SEC-Z512MGX 100 SCCM, Horiba STEC Co., Ltd., Japan) was introduced into a Picarro G2301
- 24 gas analyzer (Picarro, Inc., California, USA) at a flow rate of 0.080 L min<sup>-1</sup>, and the other (SEC-Z512MGX

**Figure 1.** Schematic diagram of the piping used to introduce the CO<sub>2</sub>-in-air mixture in a cylinder to a Picarro G2301 in the decanting experiment. MFC, mass flow controller.

1 SLM or 10 SLM, Horiba STEC Co., Ltd., Japan) was exhausted to the surroundings at flow rates of 0.0 L min<sup>-1</sup>, 0.070 L min<sup>-1</sup>, 0.22 L min<sup>-1</sup>, 1.12 L min<sup>-1</sup>, and 5.92 L min<sup>-1</sup>. An absolute pressure gauge of flush diaphragm type (PPA-33X, KELLER AG, Switzerland) attached to the regulator was used to measure pressures in the cylinders. The CO<sub>2</sub> molar fraction in this study was determined using a single-point calibration method, based on the relationship between measured values and gravimetric values of gravimetrically prepared standard gas mixtures in 10 L aluminum cylinders. For each calibration, one of nine standard gas mixtures was selected according to the measurement conditions. The gravimetric CO<sub>2</sub> molar fractions in these standard gas mixtures were from 337 μmol mol<sup>-1</sup> to 452 μmol mol<sup>-1</sup>, with standard uncertainties of less than 0.05 μmol mol<sup>-1</sup>. CO<sub>2</sub> molar fraction measurements were performed using the Picarro G2301. The standard gas mixtures were prepared by mixing pure CO<sub>2</sub> and purified air in a 10 L cylinder (Aoki et al., 2022). First, the evacuated 10 L cylinder was weighed. Pure CO<sub>2</sub> was then transferred from a 0.9 L aluminum cylinder (Luxfer Gas Cylinders, UK) into the 10 L cylinder. Subsequently, purified air was directly introduced into the same cylinder, which was weighed again after both gases had been added. The amount of CO<sub>2</sub> was determined from the mass difference of the 0.9 L cylinder before and after the transfer, using a balance (AX2005, Mettler Toledo, Switzerland) with a resolution of 0.01 mg and a maximum load of 2 kg. The amount of purified air was calculated by subtracting the CO2 mass from the

total mass increase of the 10 L cylinder following the addition of both gases. The 10 L cylinder was weighed using a separate balance (XP26003L, Mettler Toledo, Switzerland) with a resolution of 1 mg and a maximum load of 26 kg (Matsumoto et al., 2004; Aoki et al., 2019). The outflowing standard gas mixture from a cylinder with a flow rate of 0.080 L min<sup>-1</sup> was introduced directly into the Picarro G2301. After measuring the outflowing standard gas mixture cylinder for 20 min to calibrate the Picarro G2301, the outflowing gas from horizontally or vertically positioned cylinders were measured continuously for 100 min. This cycle was repeated until the pressure dropped to 0.3 MPa. In the decanting experiment at an outflowing gas rate of 6.0 L min<sup>-1</sup>, the temperatures at the top, middle, and bottom of the cylinders were measured by using a thermocouple-type thermometer that consisted of an insulated thermocouple wire (TT-K-36-SLE-100, OMEGA, Norwalk, California, USA) and a digital multimeter (DMM6500, KEITHLEY, Ohio, USA) with a scanner card (Model 2000-SCAN, KEITHLEY, Ohio, USA) as shown in Fig.1. To investigate the dependence on initial pressure, some decanting experiments were also performed at an outflowing gas flow rate of 0.15 L min<sup>-1</sup> and initial pressures of 2.1 MPa, 6.5 MPa, and 11.0 MPa.

# 2.1.3 Measurement for validation

Three experiments were conducted to validate the fractionation factors obtained by the decanting experiments. The first experiment was measurement of the deviation of the CO<sub>2</sub> value using the Picarro G2301 when the flow rate of gas leaving a cylinder was changed at 20 min intervals. Flow rates of 0.080 L min<sup>-1</sup>, 0.15 L min<sup>-1</sup>, 0.30 L min<sup>-1</sup>, 1.2 L min<sup>-1</sup>, and 6.0 L min<sup>-1</sup> were used in this experiment. The second experiment was measurement of the CO<sub>2</sub> molar fraction in outflowing gas from a cylinder positioned vertically and horizontally using the Picarro G2301 and evaluation of the difference in the CO<sub>2</sub> molar fraction between the two positions. An outflowing gas flow rate of 0.080 L min<sup>-1</sup> was used in this experiment. The third experiment was measurement of the  $\delta(\text{CO}_2/\text{N}_2)$ ,  $\delta(^{29}\text{N}_2/^{28}\text{N}_2)$ ,  $\delta(^{34}\text{O}_2/^{32}\text{O}_2)$ ,  $\delta(^{40}\text{Ar}/^{36}\text{Ar})$ ,  $\delta(^{32}\text{O}_2/^{28}\text{N}_2)$ , and  $\delta(^{40}\text{Ar}/^{28}\text{N}_2)$  at the start and end of the decanting experiment using a mass spectrometer (Delta-V, Thermo Fisher Scientific Inc., Massachusetts, USA) to clarify the contribution of thermal fractionation during the decanting experiment based on the relationship between the measured

- 1 elemental and isotopic ratios (e.g., Langenfelds et al., 2003; Ishidoya et al., 2013). The details of the
- 2 measurement technique using the mass spectrometer have been provided by Ishidoya and Murayama (2014).
- The value of  $\delta(\text{CO}_2/\text{N}_2)$  was calculated using the ratio of  $\text{CO}_2/\text{N}_2$  obtained from Eq. (1) in Aoki et al. (2022).
- 4 Here CO<sub>2</sub> molar fractions measured using Picarro G2301 were used. The ratios of O<sub>2</sub>/N<sub>2</sub> and Ar/N<sub>2</sub> were
- 5 computed using the values measured by the mass spectrometer (Aoki et al., 2019).

7

10

11

12

# 2.2 Analytical method for the decanting experiments

# 8 2.2.1 Langmuir adsorption/desorption model

9 To evaluate the deviation of the CO<sub>2</sub> molar fraction in the CO<sub>2</sub>-in-air mixture caused by

adsorption/desorption effects, the decanting experiments were repeated using vertically positioned

cylinders with low flow rates (<0.30 L min<sup>-1</sup>). Each measurement run of every cylinder was used to

individually fit a function based on the Langmuir adsorption/desorption model (Langmuir, 1916, 1918) as

derived by Leuenberger et al. (2015):

$$X_{\text{CO}_2,\text{meas}} = X_{\text{CO}_2,\text{ad}} \cdot \left( \frac{K \cdot (P - P_0)}{1 + K \cdot P} + (1 + K \cdot P_0) \cdot \ln \left( \frac{P_0 \cdot (1 + K \cdot P)}{P \cdot (1 + K \cdot P_0)} \right) \right) + X_{\text{CO}_2,\text{initial}},$$
 (1)

Where P is the actual pressure of the cylinder (MPa),  $P_0$  is the initial pressure of the cylinder (MPa) before

the decanting experiment,  $X_{CO_2,meas}$  is the measured  $CO_2$  molar fraction in the outflowing gas,  $X_{CO_2,ad}$  is

the CO<sub>2</sub> molar fraction multiplied by the occupied adsorption sites at pressure  $P_0$ ,  $X_{CO_2,initial}$  is the CO<sub>2</sub>

molar fraction measured in the outflowing gas at pressure  $P_0$ , and K is the ratio of the adsorption rate

constant to the desorption rate constant (unit MPa<sup>-1</sup>).  $X_{CO_2,ad}$ ,  $X_{CO_2,initial}$ , and K were obtained from the

nonlinear least-squares fit to the measurement results.

# 1 2.2.1 Rayleigh distillation model and its combination with the Langmuir adsorption/desorption

- 2 model
- 3 The offset of the CO<sub>2</sub> molar fraction in the outflowing gas caused by thermal diffusion fractionation can be
- 4 represented using a Rayleigh distillation model (Rayleigh, 1902; Matsubaya and Matsuo, 1982;
- 5 Langenfelds et al., 2005) according to the following equation:

$$7 \qquad \frac{X}{X_0} = \left(\frac{P}{P_0}\right)^{\alpha - 1},\tag{2}$$

where X corresponds to the measured  $CO_2$  molar fraction;  $X_0$  corresponds to the initial  $CO_2$  molar fraction in the outflowing gas; and  $\alpha$  is the fractionation factor of  $CO_2$  when the  $CO_2$ -in-air mixture leaves the cylinder. The  $CO_2$  molar fraction in the outflowing gas is depleted if  $\alpha < 1$ , which increases the  $CO_2$  molar fraction in the remaining  $CO_2$ -in-air mixture in the cylinder (and vice versa). It is possible to obtain reasonable fits to the measured  $CO_2$  molar fraction data by the Langmuir adsorption/desorption model (Eq. (1)) or Rayleigh distillation function (Eq. (2)); in other words, it is difficult to separate the contributions of adsorption/desorption and thermal diffusion fractionation. Therefore, the Langmuir–Rayleigh model, which integrates the Langmuir model and the Rayleigh function, is required to evaluate adsorption/desorption and thermal diffusion effects. The Langmuir–Rayleigh model was proposed by Schibig et al. (2018) to analyze the results of decanting experiments as follows:

$$20 X_{\text{CO}_2, meas} = X_{\text{CO}_2, ad, ave} \cdot \left( \frac{K_{ave} \cdot (P - P_0)}{1 + K_{ave} \cdot P} + (1 + K_{ave} \cdot P_0) \cdot \ln \left( \frac{P_0 \cdot (1 + K_{ave} \cdot P)}{P \cdot (1 + K_{ave} \cdot P_0)} \right) \right) + X_0 \cdot \left( \frac{P}{P_0} \right)^{\alpha - 1}, (3)$$

where  $X_{\text{CO}_2, ad, ave}$  is the average  $X_{\text{CO}_2, ad}$  coefficient of the low-flow experiments, and  $K_{ave}$  is the average ratio of the adsorption and desorption rate constants of the low-flow experiments. The value of  $\alpha$  can be obtained by fitting Eq. (3) to the results of the decanting experiments, with the values of  $X_{\text{CO}_2, ad, ave}$  and  $K_{ave}$  determined in advance.

#### 3 Results

#### 3.1 Decanting experiments

The decanting experiments were performed to evaluate the deviation of the CO<sub>2</sub> molar fraction in the 10 L aluminum cylinders resulting from thermal diffusion fractionation as the pressure dropped. Decanting the CO<sub>2</sub>-in-air mixtures from the 10 L aluminum cylinders reduced cylinder temperatures by a maximum of ~6 K depending on the outflowing gas flow rate. The temperature distribution in the cylinder depends on the outflowing gas flow rate and the cylinder position (Schibig et al., 2018; Aoki et al., 2022). The temperature reduction could also alter the amount of CO<sub>2</sub> adsorbed on the inner surface of the cylinder, because the adsorption energy changes depending on the cylinder temperature. However, the change of the adsorbed CO<sub>2</sub> amount resulting from temperature variation is estimated to be less than 0.002 μmol mol<sup>-1</sup> because the temperature dependence that was observed for aluminum cylinders by Leuenberger et al. (2015) was between -0.0002 μmol mol<sup>-1</sup> K<sup>-1</sup> and -0.0003 μmol mol<sup>-1</sup> K<sup>-1</sup>. The change is negligible because the contribution is below the CO<sub>2</sub> value reproducibility of 0.005 μmol mol<sup>-1</sup>. Therefore, CO<sub>2</sub> dilution and enrichment in cylinders with different flow rates, which ranged from -0.08 μmol mol<sup>-1</sup> to 0.31 μmol mol<sup>-1</sup> (Fig. 2), depends on thermal diffusion fractionation rather than adsorption/desorption.

# 3.1.1 Flow rate dependency

The decanting experiments were performed at outflowing gas flow rates of  $0.080 \text{ L min}^{-1}$ ,  $0.15 \text{ L min}^{-1}$ ,  $0.30 \text{ L min}^{-1}$ ,  $1.2 \text{ L min}^{-1}$  and  $6.0 \text{ L min}^{-1}$  for cylinders positioned horizontally and vertically until the pressure dropped from 10 MPa to 0.3 MPa. Figure 2 shows the deviations of the CO<sub>2</sub> molar fraction in the outflowing gas as the relative pressure  $(P/P_0)$  in the cylinders dropped. For a horizontally positioned cylinder, the deviations of CO<sub>2</sub> molar fraction at a relative pressure of 0.03 were between  $0.06 \text{ }\mu\text{mol mol}^{-1}$  to  $-0.08 \text{ }\mu\text{mol mol}^{-1}$  relative to the initial CO<sub>2</sub> molar fractions as summarized in Table 1. The deviation decreased as the flow rate increased, indicating that thermal diffusion fractionation acted to dilute the CO<sub>2</sub>

**Figure 2.** Plot showing deviation of the CO<sub>2</sub> molar fraction from the initial value against relative pressure. These results were obtained by decanting experiments at outflowing gas flow rates between 0.080 L min<sup>-1</sup>, 0.15 L min<sup>-1</sup>, 0.30 L min<sup>-1</sup>, 1.2 L min<sup>-1</sup>, and 6.0 L min<sup>-1</sup> with vertically positioned cylinders and horizontally positioned cylinders

molar fraction in the horizontally positioned cylinder because adsorption/desorption acted to enrich the CO<sub>2</sub> molar fraction (Leuenberger et al., 2015; Schibig et al., 2018; Aoki et al., 2022). These results also mean that the contribution of thermal diffusion fractionation increased at higher flow rates. At a flow rate of 0.080 L min<sup>-1</sup>, the CO<sub>2</sub> molar fraction was enriched as the relative pressure dropped, indicating that the effect of adsorption/desorption was larger than that of thermal diffusion fractionation. At flow rates of 0.15 L min<sup>-1</sup> and 0.30 L min<sup>-1</sup>, the CO<sub>2</sub> molar fractions were almost constant, indicating that the increase due to adsorption/desorption was cancelled out by the decrease due to thermal diffusion fractionation. At flow

- 1 rates of 1.2 L min<sup>-1</sup> and 6.0 L min<sup>-1</sup>, the CO<sub>2</sub> molar fractions decreased as the pressure dropped, indicating
- 2 that the thermal diffusion effect was larger than the adsorption/desorption effect.

**Table 1** Deviations of CO<sub>2</sub> molar fraction in the outflowing gas from initial values measured by decanting experiments at flow rates of 0.080 L min<sup>-1</sup>, 0.15 L min<sup>-1</sup>, 0.30 L min<sup>-1</sup>, 1.2 L min<sup>-1</sup>, and 6.0 L min<sup>-1</sup>.

| Flow rate                  | Deviations at a relative pressure of 0.03 (μmol mol <sup>-1</sup> ) |                                |  |  |
|----------------------------|---------------------------------------------------------------------|--------------------------------|--|--|
|                            | Horizontally positioned cylinder                                    | Vertically positioned cylinder |  |  |
| 0.080 L min <sup>-1</sup>  | 0.06                                                                | 0.12                           |  |  |
| $0.15~\mathrm{L~min^{-1}}$ | -0.002                                                              | 0.11                           |  |  |
| $0.30~\mathrm{L~min^{-1}}$ | -0.005                                                              | 0.12                           |  |  |
| 1.2 L min <sup>-1</sup>    | -0.08                                                               | 0.20                           |  |  |
| 6.0 L min <sup>-1</sup>    | -0.08                                                               | 0.31                           |  |  |

For vertically positioned cylinders, at all outflow rates the CO2 molar fraction in the outflowing gas

increased from the initial value as the pressure dropped. The increases in the CO<sub>2</sub> molar fractions at a relative pressure of 0.03 were between 0.12 µmol mol<sup>-1</sup> and 0.31 µmol mol<sup>-1</sup> relative to the initial values (Table 1). The increases were larger at higher flow rates, indicating that thermal diffusion fractionation acted to enrich the CO<sub>2</sub> molar fraction and its contribution was greater with increased flow rate. However, there was little difference in the CO<sub>2</sub> enrichment for flow rates less than 0.30 L min<sup>-1</sup>, suggesting that the contribution of thermal diffusion fractionation was minimal at these rates, and the CO<sub>2</sub> enrichment can mainly be attributed to adsorption/desorption effects.

To understand the mechanism of thermal diffusion fractionation, the temperatures at the top, middle, and bottom of the cylinders were measured using a thermocouple-type thermometer (Fig.1). Figure 3a shows the relationship of pressure and temperatures at the top, middle, and bottom of a horizontally positioned cylinder when decanting the CO<sub>2</sub>-in-air mixture from 10 MPa to 0.3 MPa at a flow rate of 6.0 L min<sup>-1</sup>. The temperatures at the top, middle, and bottom of the cylinder decreased as the pressure dropped, while the temperatures at the top, middle, and bottom were almost equivalent at all pressures. These results do not provide insights into the thermal distribution that drives thermal diffusion fractionation; thus, further study of the mechanism of thermal diffusion fractionation in a horizontally positioned cylinder is required. Figure

**Figure 3.** Temperature changes from the initial values of at the top, middle, and bottom of the cylinder when the CO<sub>2</sub>-in-air mixture was decanted at a flow rate of 6 L min<sup>-1</sup>.

3b shows the relationship between pressure and the temperatures at the top, middle, and bottom of a vertically positioned cylinder during decanting of the CO<sub>2</sub>-in-air mixture at a flow rate of 6.0 L min<sup>-1</sup> from 10 MPa to 0.3 MPa. The temperatures at the top, middle, and bottom of the cylinder decreased as the pressure dropped, reaching stable values below 2 MPa, while the temperature differences between the different parts of the cylinder increased as the pressure dropped. The temperature difference between the

- 1 cylinder top and bottom was about 0.7 °C at pressures below 2 MPa, indicating that thermal diffusion
- 2 fractionation was caused by the temperature difference between the upper and lower parts of the cylinder.

#### 3.1.2 Evaluation of adsorption/desorption effect

3

4 CO<sub>2</sub> enrichment in a vertically positioned cylinder is considered to be mainly due to adsorption/desorption 5 in the decanting experiment performed at an outflowing gas flow rate of less than 0.30 L min<sup>-1</sup>, as described 6 in Sect. 3.1.1. In this section, the adsorption/desorption effect was quantitively evaluated from the results obtained by repeating the decanting experiment at a flow rate of less than 0.30 L min<sup>-1</sup> with a vertically 7 8 positioned cylinder. In this experiment, the CO2 enrichment was assumed to be caused by only 9 adsorption/desorption effects. 10 The decanting experiments were initially repeated seven times with a CPC00494 cylinder to determine the 11 measurement uncertainty of CO<sub>2</sub> enrichment as the pressure dropped. The Langmuir model was fitted to 12 each measurement result. The average values of K and  $X_{\rm CO_2,ad}$  were  $0.020 \pm 0.036~\rm MPa^{-1}$  and  $0.027 \pm$ 13 0.002 μmol mol<sup>-1</sup>, respectively. Here, the number following the symbol represents the standard deviation. 14 The decanting experiments were then repeated 10 times, each with a different cylinder with the same types 15 of internal surface treatment and diaphragm valve, to determine the adsorption/desorption effect 16 quantitatively. Figure 4a shows the deviations of the CO2 molar fraction in the outflowing gas with 17 decreasing P/P<sub>0</sub>, obtained from the decanting experiments with 10 replicates. The CO<sub>2</sub> molar fraction increased from  $0.08 \,\mu\text{mol mol}^{-1}$  to  $0.15 \,\mu\text{mol mol}^{-1}$  from initial values as  $P/P_0$  dropped to 0.03. The average 18 19 K and  $X_{\text{CO}_2,\text{ad}}$  values were  $0.024 \pm 0.035 \text{ MPa}^{-1}$  and  $0.028 \pm 0.005 \text{ }\mu\text{mol mol}^{-1}$ , respectively, when fitting 20 a function based on the Langmuir model. The averages were consistent with that for the CPC00494 cylinder 21 within uncertainty, demonstrating that K and  $X_{CO_2,ad}$  do not differ in different cylinders. 22 In addition, decanting experiments were performed at initial pressures P<sub>0</sub> of 2.1 MPa, 6.5 MPa, and 11.0 MPa with a vertically positioned cylinder. A flow rate of 0.15 L min<sup>-1</sup> was used for the outflowing gas. 23 24 Figure 4b shows the deviations of the CO<sub>2</sub> molar fraction in the outflowing gas from the CO<sub>2</sub>-in-air mixture 25 with decreasing  $P/P_0$ . The deviations obtained from the three experiments agreed well with each other, indicating that the adsorption/desorption effect in the vertically positioned cylinder depends on  $P/P_0$  rather

# 2 than P.

**Figure 4.** (a) Deviations of the CO<sub>2</sub> molar fraction from the initial value versus relative pressure at flow rates of less than 0.30 L min<sup>-1</sup> in vertically positioned cylinders. (b) Deviations of the CO<sub>2</sub> molar fraction from the initial value versus relative pressure for initial pressures of 2.1 MPa, 6.5 MPa, and 11.0 MPa.

**Figure 5.** Results from fitting the equation combining the Langmuir and Rayleigh distillation functions to the deviations of the CO<sub>2</sub> molar fraction versus relative pressure. (**a–e**) Results for horizontally positioned cylinders. (**f–j**) Results for vertically positioned cylinders.

#### 3.1.3 Estimation of thermal diffusion fractionation

1

- 2 Fractionation factors for the CO<sub>2</sub>-in-air mixture leaving the cylinders were obtained by fitting a function
- 3 based on the Langmuir-Rayleigh model (Eq. (3)) to the results described in Sect. 3.1.1; the functions are

**Table 2** Fractionation factors for CO<sub>2</sub>-in-air mixtures leaving 10 L aluminum cylinders obtained by fitting the Langmuir–Rayleigh model to the decanting measurements. Offsets and differences are from the original values in the cylinders and from the values for 0.080 L min<sup>-1</sup>, and they were calculated from the fractionation factors.

| Flow rate                        | Fractionation factor <sup>a</sup> |                           |                                       | Theoretical differences   |  |
|----------------------------------|-----------------------------------|---------------------------|---------------------------------------|---------------------------|--|
|                                  |                                   |                           | from the 0.080 L                      |                           |  |
|                                  |                                   | values b                  | min <sup>-1</sup> values <sup>c</sup> | values d                  |  |
|                                  |                                   | (µmol mol <sup>-1</sup> ) | (μmol mol <sup>-1</sup> )             | (μmol mol <sup>-1</sup> ) |  |
| Horizontally positioned cylinder |                                   |                           |                                       |                           |  |
| $0.080~L~min^{-1}$               | $1.000041 \pm 0.000001$           | $0.017\pm0.000$           | _                                     | _                         |  |
| $0.15~L~min^{-1}$                | $1.000082 \pm 0.000001$           | $0.034\pm0.000$           | $0.009 \pm 0.018$                     | $0.017 \pm 0.001$         |  |
| $0.30\;L\;min^{-1}$              | $1.000095 \pm 0.000002$           | $0.040\pm0.001$           | $0.025 \pm 0.018$                     | $0.023 \pm 0.001$         |  |
| 1.2 L min <sup>-1</sup>          | $1.000150 \pm 0.000005$           | $0.063\pm0.002$           | $0.049 \pm 0.018$                     | $0.046\pm0.002$           |  |
| $6.0~L~min^{-1}$                 | $1.000164 \pm 0.000007$           | $0.069\pm0.003$           | $0.050\pm0.018$                       | $0.052 \pm 0.003$         |  |
| Vertically positioned cylinder   |                                   |                           |                                       |                           |  |
| $0.080\;L\;min^{-1}$             | $1.000000 \pm 0.000001$           | $0.000\pm0.001$           | _                                     | _                         |  |
| $0.15~L~min^{-1}$                | $1.000007 \pm 0.000002$           | $0.003\pm0.001$           | $0.010\pm0.018$                       | $0.003 \pm 0.001$         |  |
| $0.30\;L\;min^{-1}$              | $1.000002 \pm 0.000002$           | $0.001\pm0.001$           | $0.011 \pm 0.018$                     | $0.001 \pm 0.001$         |  |
| 1.2 L min <sup>-1</sup>          | $0.999938 \pm 0.000005$           | $-0.026 \pm 0.002$        | $-0.016 \pm 0.018$                    | $-0.025 \pm 0.002$        |  |
| $6.0~L~min^{-1}$                 | $0.999852 \pm 0.000005$           | $-0.062 \pm 0.002$        | $-0.074 \pm 0.018$                    | $-0.060 \pm 0.002$        |  |

The number following the symbol represents the standard uncertainty.

<sup>&</sup>lt;sup>a</sup> These values were calculated by fitting Eq. (3) to the results of the decanting experiments described in Sect. 3.1.1. The standard uncertainty represents the standard deviation obtained from the fitting.

<sup>&</sup>lt;sup>b</sup> Offsets of the CO<sub>2</sub> molar fraction in the outflowing gas for a CO<sub>2</sub>-in-air mixture with an original molar fraction of 420 μmol mol<sup>-1</sup>. These offsets represent the differences between the original values and the values obtained by multiplying the original values by the fractionation factors.

<sup>&</sup>lt;sup>c</sup> Differences from the  $0.080\,L$  min<sup>-1</sup> value when changing the outflowing gas flow rate. These differences were determined by measuring CO<sub>2</sub>-in-air mixtures with CO<sub>2</sub> molar fractions of 421.2 μmol mol<sup>-1</sup> and 406.5 μmol mol<sup>-1</sup> for horizontally and vertically positioned cylinders, respectively.

<sup>&</sup>lt;sup>d</sup> Differences from the  $0.080\,L$  min<sup>-1</sup> value when changing the outflowing gas flow rate when decanting CO<sub>2</sub>-in-air mixtures with CO<sub>2</sub> molar fractions of 421.2 μmol mol<sup>-1</sup> and 406.5 μmol mol<sup>-1</sup> for horizontally and vertically positioned cylinders, respectively. These differences were calculated based on the fractionation factors.

- shown in Fig. 5. The constant coefficient  $K_{ave}$  and  $X_{CO_2, ad, ave}$  were  $0.024 \pm 0.035 \text{ MPa}^{-1}$  and  $0.028 \pm$
- 0.005  $\mu$ mol mol<sup>-1</sup>, respectively, as determined in the previous section. The fractionation factors ( $\alpha$ )
- obtained from the fit functions following the Langmuir–Rayleigh model and the deviation of the CO<sub>2</sub> molar
- fraction calculated based on the  $\alpha$  values are summarized in Table 2.
- For horizontally positioned cylinders, the fractionation factor  $\alpha$  was between 1.000041  $\pm$  0.000001 and
- 1.000164  $\pm$  0.000007; the outflowing gas had offsets between  $0.017 \pm 0.000 \,\mu\text{mol mol}^{-1}$  and  $0.069 \pm 0.002$
- 7 μmol mol<sup>-1</sup> from the original values in the cylinders (Table 2) for a CO<sub>2</sub> molar fraction of 420 μmol mol<sup>-1</sup>
- (the atmospheric level).
- For vertically positioned cylinders,  $\alpha$  was between  $1.000000 \pm 0.000001$  and  $0.999852 \pm 0.000005$ ; the
- outflowing gas had offsets between  $0.000 \pm 0.001$  µmol mol<sup>-1</sup> and  $-0.062 \pm 0.002$  µmol mol<sup>-1</sup> from the
- original values, respectively (Table 2) for a  $CO_2$  molar fraction of 420  $\mu$ mol mol<sup>-1</sup>.

# 12 3.2 Validation of thermal diffusion fractionation

- 13 The fractionation factors determined in the previous section were validated in three ways: first, by
- measuring the offset of the CO<sub>2</sub> molar fraction corresponding to the fractionation factors when changing
- 15 the flow rate of the outflowing gas (see Sect. 3.2.1); second, by measuring the CO<sub>2</sub> molar fraction in the
- outflowing gas from the same horizontally or vertically positioned cylinder at a flow rate of 0.080 L min<sup>-1</sup>
- 17 and comparing the difference between values (see Sect. 3.2.2); and third, by measuring δ(CO<sub>2</sub>/N<sub>2</sub>),
- 18  $\delta(^{40}\text{Ar}/^{36}\text{Ar})$ ,  $\delta(^{34}\text{O}_2/^{32}\text{O}_2)$ ,  $\delta(^{40}\text{Ar}/^{28}\text{N}_2)$ ,  $\delta(^{32}\text{O}_2/^{28}\text{N}_2)$ , and  $\delta(^{40}\text{Ar}/^{36}\text{Ar})$  by mass spectrometry before and
- after the decanting experiment (see Sect. 3.2.3).

20

#### 3.2.1 Deviations of CO<sub>2</sub> molar fractions at different flow rates

- 21 The fractionation factors determined in Sect. 3.1.3 suggest that the CO<sub>2</sub> molar fractions in the outflowing
- 22 gas have the offsets from the original values depending on the flow rate. The outflowing gas from
- 23 horizontally and vertically positioned cylinders with CO<sub>2</sub> molar fractions of 421.2 μmol mol<sup>-1</sup> and 406.6

- 1 μmol mol<sup>-1</sup> were continuously measured as the outflowing gas flow rate was varied from 0.080 L min<sup>-1</sup> to
- 2 6.0 L min<sup>-1</sup> at 20 min intervals.
- The differences from the CO<sub>2</sub> value of the  $0.080 \text{ L min}^{-1}$  flow rate were between  $0.009 \pm 0.018 \text{ }\mu\text{mol mol}^{-1}$
- 4 and  $0.050 \pm 0.018 \,\mu\text{mol mol}^{-1}$  (Table 2). Here, the number following the symbol represents the standard
- 5 uncertainty ( $\sqrt{0.013^2+0.013^2}=0.018$ ), which was calculated by combining the measurement repeatability
- of the CO<sub>2</sub> values (0.013 μmol mol<sup>-1</sup>) at each flow rate and at 0.080 L min<sup>-1</sup>. The theoretical differences
- 7 from the CO<sub>2</sub> value at 0.080 L min<sup>-1</sup> were calculated based on the fractionation factors to be between 0.017
- $8 \pm 0.001 \mu mol mol^{-1}$  and  $0.052 \pm 0.003 \mu mol mol^{-1}$  (Table 2). The measured difference values agreed with
- 9 the theoretical values within the uncertainties, suggesting that the differences between the fractionation
- factors are valid for the horizontally positioned cylinders.
- The differences from the CO<sub>2</sub> value at 0.080 L min<sup>-1</sup> were between  $0.010 \pm 0.018 \,\mu\text{mol mol}^{-1}$  and -0.074
- $\pm 0.018$  μmol mol<sup>-1</sup>. Theoretical differences from the CO<sub>2</sub> value for 0.080 L min<sup>-1</sup> were calculated based
- on the fractionation factors to be between  $0.003 \pm 0.001 \,\mu\text{mol} \,\text{mol}^{-1}$  and  $-0.060 \pm 0.002 \,\mu\text{mol} \,\text{mol}^{-1}$  (Table
- 14 2). All of the measured difference values also agreed with the theoretical values, suggesting that the
- differences in fractionation factor are valid for vertically positioned cylinders.

# 3.2.2 Difference in CO<sub>2</sub> molar fractions for vertically and horizontally positioned cylinders

- The dependence of the fractionation factor on the outflowing gas flow rates in each cylinder position was
- verified in the previous section; however, the difference between vertically and horizontally positioned
- cylinders was not verified. In this section, the CO<sub>2</sub> differences of a cylinder containing the same CO<sub>2</sub>-in-air
- mixture with a CO<sub>2</sub> molar fraction of 391.9 μmol mol<sup>-1</sup> was measured in both horizontal and vertical
- positions to evaluate whether an offset of the CO<sub>2</sub> molar fraction corresponding to the fractionation factors
- could be detected between the positions.

- The CO<sub>2</sub> offsets at an outflow rate of 0.080 L min<sup>-1</sup> were calculated to be 0.017  $\pm$  0.001  $\mu$ mol mol<sup>-1</sup>
- 24 (horizontal cylinder) and  $0.000 \pm 0.001 \,\mu\text{mol mol}^{-1}$  (vertical cylinder), based on fractionation factors of

- $1.000041 \pm 0.000001$  (horizontal cylinder) and  $1.000000 \pm 0.000001$  (vertical cylinder). The
- 2 difference of the CO<sub>2</sub> molar fraction between the horizontal and vertical positions is estimated to be 0.017
- $3 \pm 0.001 \mu mol^{-1}$ . Here, the number following the symbol is the standard uncertainty obtained by
- 4 combining the uncertainties of both offsets. To detect the difference, the cylinder was left in a horizontal

**Figure 6.** CO<sub>2</sub> molar fraction in a cylinder measured in both vertical and horizontal positions. Error bars represent standard errors.

position overnight and measured once, then left in a vertical position overnight and measured once, and the measurement sequence was performed four times. The average value of the measured difference between the two positions was  $0.011 \pm 0.004~\mu mol~mol^{-1}$  (Fig. 6). The number following the symbol represents the standard uncertainty, which was calculated by combining the standard error of the  $CO_2$  molar fraction for each cylinder position. The expanded uncertainties (k=2) of the measured and estimated differences were  $0.008~\mu mol~mol^{-1}$  and  $0.002~\mu mol~mol^{-1}$ , respectively. These measured and estimated differences of  $0.011~\pm~0.008~\mu mol~mol^{-1}$  and  $0.017~\pm~0.002~\mu mol~mol^{-1}$  are in agreement within uncertainty, suggesting that the difference in the fractionation factors is valid between horizontal and vertical cylinder positions.

# 3.2.3 Contribution of thermal diffusion fractionation at 0.080 L min<sup>-1</sup> flow rate

1

2 As discussed above, the relationship of the fractionation factors between the different outflowing gas rates 3 summarized in Table 2 is relatively valid. However, the fractionation factors were calculated by assuming 4 that thermal diffusion fractionation was negligible for gas flowing out from a vertically positioned cylinder at a flow rate of less than  $0.30 \text{ L min}^{-1}$  (Fig. 2). To validate this assumption, we measured  $\delta(\text{CO}_2/\text{N}_2)$ , 5  $\delta(^{40}\text{Ar}/^{36}\text{Ar}), \delta(^{34}\text{O}_2/^{32}\text{O}_2), \delta(^{40}\text{Ar}/^{28}\text{N}_2), \delta(^{32}\text{O}_2/^{28}\text{N}_2), \text{ and } \delta(^{29}\text{N}_2/^{28}\text{N}_2) \text{ in the outflowing gas before and after}$ 6 7 decanting from 8 MPa to below 0.9 MPa. The experiments were carried out by using a vertically positioned 8 cylinder with flow rates of 0.080 L min<sup>-1</sup>, 0.15 L min<sup>-1</sup>, and 0.30 L min<sup>-1</sup>, and a horizontally positioned 9 cylinder with a flow rate of 0.080 L min<sup>-1</sup>. 10 Figure 7 shows the relationship of the deviations of  $\delta(\text{CO}_2/\text{N}_2)$ ,  $\delta(^{40}\text{Ar}/^{36}\text{Ar})$ ,  $\delta(^{34}\text{O}_2/^{32}\text{O}_2)$ ,  $\delta(^{40}\text{Ar}/^{28}\text{N}_2)$ , and 11  $\delta(^{32}O_2/^{28}N_2)$  values against those of  $\delta(^{29}N_2/^{28}N_2)$ . The deviations of  $\delta(CO_2/N_2)$  observed in this study are 12 generally larger than the dotted line, whereas most of the  $\delta(\text{CO}_2/\text{N}_2)$  values reported in Aoki et al. (2022), 13 which were primarily influenced by thermal diffusion fractionation, agree with the dotted line within their 14 respective uncertainty. The difference from the dotted line indicates an additional deviation attributable to adsorption effect. In contrast, the deviations of  $\delta(^{40}\text{Ar}/^{36}\text{Ar})$ ,  $\delta(^{34}\text{O}_2/^{32}\text{O}_2)$ ,  $\delta(^{40}\text{Ar}/^{28}\text{N}_2)$ , and  $\delta(^{32}\text{O}_2/^{28}\text{N}_2)$ 15 16 values against  $\delta(^{29}N_2/^{28}N_2)$  mostly fall on the dotted lines within uncertainties, suggesting that the observed 17 negative and positive deviations for horizontal and vertical cylinders were caused by thermal diffusion 18 fractionation. Thus, thermal diffusion fractionation occurs even at low flow rates, regardless of whether the 19 cylinders are positioned horizontally or vertically. Furthermore, the consistent patterns observed in  $\delta(^{40}\text{Ar}/^{36}\text{Ar}), \delta(^{34}\text{O}_2/^{32}\text{O}_2), \delta(^{40}\text{Ar}/^{28}\text{N}_2), \text{ and } \delta(^{32}\text{O}_2/^{28}\text{N}_2) \text{ support that the CO}_2 \text{ values in the horizontal and}$ 20 21 vertical cylinders also deviate negatively and positively by thermal diffusion fractionation. The deviations of  $\delta(^{29}N_2/^{28}N_2)$  at a flow rate of 0.080 L min<sup>-1</sup> were  $-2.7 \pm 1.4$  per meg in the depletion from 22 23 8.3 to 0.6 MPa for the horizontally positioned cylinder and  $3.9 \pm 1.4$  per meg in the depletion from 8.5 MPa 24 to 0.2 MPa for the vertically positioned cylinder. These values correspond to  $CO_2$  deviations of  $-0.032 \pm$ 

**Figure 7.** Relationship between the deviations of  $\delta(^{34}O_2/^{32}O_2)$ ,  $\delta(^{40}Ar/^{36}Ar)$ ,  $\delta(^{32}O_2/^{28}N_2)$ , and  $\delta(^{40}Ar/^{28}N_2)$  and those of  $\delta(^{29}N_2/^{28}N_2)$  in daughter cylinders relative to their initial value when  $CO_2/air$  mixtures with an atmospheric  $CO_2$  level were decanted from the cylinder. The error bars indicate the expanded uncertainties (k = 2) of the deviations. The dotted lines represent the deviations due to thermal diffusion, which were experimentally estimated by Ishidoya et al. (2013, 2014). The black closed circles represent the deviations in daughter cylinders relative to their mother cylinders obtained by mother—daughter experiments (Aoki et al. 2022). However, the  $CO_2/N_2$  values were corrected for adsorption/desorption effect based on the values of Aoki et al. (2022).

 $0.017 \text{ }\mu\text{mol mol}^{-1}$  and  $0.047 \pm 0.017 \text{ }\mu\text{mol mol}^{-1}$  for horizontally and vertically positioned cylinders, respectively, based on the relationship between the  $\delta(\text{CO}_2/\text{N}_2)$  deviations and those of  $\delta(^{29}\text{N}_2/^{28}\text{N}_2)$  for thermal fractionation as shown in Fig. 7. The number following the symbol indicates the standard uncertainties of the deviations, which were based on the uncertainties of the deviations of  $\delta(^{29}N_2/^{28}N_2)$ . When substituting the CO<sub>2</sub> molar fractions and the pressures before and after each decanting experiments into the function based on the Rayleigh distillation model (Eq. (2)), the fractionation factors were calculated to be  $1.000030 \pm 0.000037$  for the horizontally positioned cylinder and  $0.999968 \pm 0.000027$  for the vertically positioned cylinder with the atmospheric CO<sub>2</sub> level of 420 μmol mol<sup>-1</sup>. The fractionation factors correspond to offsets in the outflowing gas of  $0.013 \pm 0.015 \,\mu\text{mol mol}^{-1}$  (horizontal cylinder) and -0.014± 0.011 µmol mol<sup>-1</sup> (vertical cylinder), meaning that the CO<sub>2</sub> molar fraction in the horizontally and vertically positioned cylinder deviated by  $-0.045 \,\mu\text{mol mol}^{-1}$  and  $0.048 \,\mu\text{mol mol}^{-1}$ , respectively, as the relative pressure dropped to 0.03. The difference in the CO<sub>2</sub> molar fraction between outflowing gases for both cylinder positions was calculated to be  $0.027 \pm 0.038 \, \mu \text{mol mol}^{-1}$ , consistent with the difference of  $0.011 \pm 0.008 \,\mu \text{mol} \,\text{mol}^{-1}$  between the horizontally and vertically positioned cylinders obtained in the previous section. The numbers after the symbol represent the expanded uncertainties (k = 2), which were calculated by combining the standard uncertainties for both cylinder positions. This finding indicates that the fractionation factors obtained using the mass spectrometer are reasonable and the assumption that thermal diffusion fractionation is negligible in the vertical position was not correct. The difference from the fractionation factor of less than 0.080 L min<sup>-1</sup> in the vertical position is reasonable, although the absolute fractionation factors need to be revised based on the fractionation factors obtained using the mass spectrometer.

### 4 Discussion

- In actual atmospheric observation, the standard gas mixture is used intermittently rather than continuously,
- whereas the results in this study are based on decanting experiments in which the CO<sub>2</sub>-in-air mixture was

used continuously. Therefore, it is necessary to confirm that adsorption and thermal diffusion effects are equivalent between continuous and intermittent use of standard gas mixtures, to be able to discuss how to operate the standard gas mixtures taking into account the results from this study. Schibig et al. (2018) reported that the  $CO_2$  desorption energy ( $E_d$ ) from an aluminum cylinder inner surface was 10 kJ mol<sup>-1</sup>, meaning that the only adsorption mechanism for  $CO_2$  on the inner wall of the cylinder is physisorption. The desorption lifetime  $\tau$  on the inner surface of the cylinder is expressed by the following Arrhenius-type equation (Arrhenius, 1889a, b; Laidler, 1949; Frenkel, 1924; Laidler et al., 1940):

$$\tau = \frac{1}{A_{des}} \times e^{Ed/RT},\tag{4}$$

where  $A_{des}$  is a pre-exponential factor (10<sup>12</sup> s) (Knopf et al., 2024), and R and T represent the gas constant  $(8.314 \text{ J K}^{-1} \text{ mol}^{-1})$  and room temperature (298 K), respectively. Using these values,  $\tau$  is calculated to be  $6\times10^{-11}$  s. Because the desorption lifetime is sufficiently shorter than the pressure change rate of  $1.4\times10^{-11}$ 10<sup>-5</sup> MPa s<sup>-1</sup>, the CO<sub>2</sub> on the inner surface and in the standard gas mixture is estimated to have always been in equilibrium over the experiments in this study. The adsorption/desorption effect would be comparable for intermittent and continuous use. However, thermal diffusion fractionation could differ between intermittent and continuous use if the thermal distribution in the cylinder takes a long time to reach equilibrium. The equilibrium time for the temperature distribution can be estimated from the time it takes for the CO<sub>2</sub> value to stabilize; in the experiment in Sect. 3.2.1, the temperature distribution reaches equilibrium within a few minutes even when the flow rate of the outflowing gas is changed. Because actual measurements of standard gas mixtures are carried out continuously over several tens of minutes, which is longer than the equilibrium time for the thermal distribution, it can be estimated that even intermittent use is not markedly different from continuous measurements. Hence, we discuss how to operate the standard gas mixtures based on the results of this study. In this study, the CO<sub>2</sub> molar fraction was determined solely from the <sup>12</sup>C<sup>16</sup>O<sup>16</sup>O signal measured by the Picarro G2301. Therefore, the reported CO<sub>2</sub> molar fraction is unlikely to be influenced by changes in the isotopic composition of CO<sub>2</sub> that may arise from adsorption/desorption and/or thermal diffusion

fractionation. Adsorption/desorption effects arise from differences in intermolecular van der Waals forces, which are governed by the electronic structure of molecules—such as their polarity and polarizability. Because isotopes (e.g., <sup>12</sup>C and <sup>13</sup>C) differ only in the number of neutrons and thus in mass, their electronic structures are essentially identical. As a result, differences in van der Waals interactions between isotopologues are negligible, and isotope fractionation associated with physical processes like adsorption is expected to be minimal. Furthermore, Sugawara et al. (2025) suggested that isotopic fractionation of  $\delta(^{13}CO_2)^{12}CO_2$ ) due to gravitational separation can be approximated by  $\delta(^{29}N_2)^{28}N_2$ ). This is because both have the same mass number difference from their respective reference isotopologues. Variations in isotopic ratios due to thermal diffusion do not show complete mass-dependent fractionation unlike gravitational separation (Severinghaus et al., 2001; Ishidoya et al., 2013). Nevertheless, the thermal diffusion sensitivities reported by Severinghaus et al. (2001) increase with increasing mass number difference and their dependences on mass number difference are smaller than those expected from gravitational separation, then 100 per meg change in  $\delta(^{29}N_2/^{28}N_2)$ (actual observed change: 7 per meg) would correspond to a 0.1% change in  $\delta(^{13}CO_2/^{12}CO_2)$ . In other words, even if a 1 ppm change in total CO<sub>2</sub> concentration were detected in this study, the corresponding change in <sup>13</sup>CO<sub>2</sub> would be only about 0.0026 ppm. Therefore, the contribution of isotopic variation to thermal diffusion fractionation-and thus to the observed CO<sub>2</sub> molar fraction-is considered negligible. Thermal diffusion fractionation has been demonstrated to have diluted the CO2 molar fraction in the horizontal cylinder and to have enriched the molar fraction in the vertical cylinder as the pressure dropped. This effect also increased as the outflowing gas flow rate increased, although the adsorption/desorption effect was constant. These findings are consistent with the results of previous studies (Schibig et al., 2018; Aoki et al., 2022). Furthermore, information on the mechanism of thermal diffusion fractionation was obtained from the temperature changes at the top, middle, and bottom of the cylinder monitored in the decanting experiments performed with horizontally and vertically positioned cylinders at a flow rate of 6 L

min<sup>-1</sup>. The temperature difference between the top, middle, and bottom of the cylinder was negligible when the cylinder was horizontal, but the temperatures at the top and bottom were 0.3 K and -0.4 K higher than that in the middle when the cylinder was vertical (Fig. 3). The offset of the outflowing gas was 0.069 µmol mol<sup>-1</sup> for the horizontal cylinder and  $-0.062 \mu \text{mol mol}^{-1}$  for the vertical cylinder (Table 2). The detected offset was estimated to be driven by a temperature difference of 0.9 K, computed using the thermal diffusion coefficient reported by Severinghaus et al. (1996). Because the thermal conductivity of the aluminum cylinder is higher than that of the internal gas, the measured temperature difference of the cylinder is expected to be smaller than the actual temperature difference of the gas mixture. The temperature difference of 0.3 K between the top and middle of the vertical cylinder appears to support the validity of the calculated temperature difference of 0.9 K. These results mean that the outflowing gas would have been taken out from the warmer gas at the cylinder top of the vertical cylinder, although the temperature distribution causing thermal diffusion fractionation could not be determined for the horizontal cylinder. However, the contribution of thermal diffusion fractionation has been understood to be negligible at low flow rates in previous studies (Schibig et al., 2018; Hall et al., 2019; Aoki et al., 2022). Notwithstanding, even at a flow rate of 0.080 L min<sup>-1</sup>, which is within the usual range of flow rates used by observation laboratories, we found that thermal diffusion fractionation produced offsets of the CO<sub>2</sub> molar fractions of  $0.013 \pm 0.015$  $\mu$ mol mol<sup>-1</sup> (horizontal cylinder) and  $-0.014 \pm 0.011 \mu$ mol mol<sup>-1</sup> (vertical cylinder) in the outflowing gases. These offsets are driven by temperature difference in the cylinder as small as 0.18 K and indicate that a difference of  $0.027 \,\mu \text{mol mol}^{-1}$  can be produced simply by changing the cylinder from horizontal to vertical. Measuring standard gas mixtures while keeping the cylinder in the same position will be an effective means of maintaining the long-term consistency of observed values, because it is difficult to completely suppress the occurrence of such small temperature differences. Furthermore, the CO<sub>2</sub> deviation resulting from the adsorption/desorption effect and thermal diffusion effects as pressure dropped were verified using the results of the decanting experiment at a flow rate of 0.080 L min<sup>-1</sup> in Fig. 2. The CO<sub>2</sub> deviation due to thermal diffusion fractionation as the pressure dropped

**Figure 8.** Total CO<sub>2</sub> deviation, deviation due to adsorption/desorption, and deviation due to thermal diffusion fractionation relative to the initial value for a CO<sub>2</sub>-in-air mixture leaving (**a**) a horizontally positioned cylinder and (**b**) a vertically positioned cylinder at an outflowing gas flow rate of 0.080 L min<sup>-1</sup>.

- 1 was calculated by substituting fractionation factors of  $1.000030 \pm 0.000037$  for a horizontally positioned
- cylinder and  $0.999968 \pm 0.000027$  for a vertically positioned cylinder into the Rayleigh function (Eq. (2))

(orange shading in Fig. 8). The total CO<sub>2</sub> deviation was estimated by fitting the Langmuir–Rayleigh model (Eq. (3)) to the results of the decanting experiment at a flow rate of 0.080 L min<sup>-1</sup>. The CO<sub>2</sub> deviation due to adsorption/desorption was calculated by subtracting the thermal diffusion fractionation deviation from the total CO<sub>2</sub> deviation. The contributions to the total change were 60% (vertical cylinder) and 70% (horizontal cylinder) for adsorption/desorption, and those of thermal diffusion fractionation were 40% (vertical cylinder) and 30% (horizontal cylinder) (Fig. 8). Here, the CO<sub>2</sub> deviation from the initial value due to adsorption/desorption at a relative pressure of 0.03 was ~0.1 μmol mol<sup>-1</sup> for a horizontally positioned cylinder and ~0.06 μmol mol<sup>-1</sup> for a vertically positioned cylinder. This difference is assumed to be uncertainty because the contribution of adsorption/desorption should be constant regardless of the cylinder position. That is, it is necessary to understand that this estimated contribution contains a large uncertainty. The contribution of adsorption/desorption is larger than that of thermal diffusion, but fractionation due to thermal diffusion is not negligible. The WMO recommends that calibration standard gas mixtures of CO<sub>2</sub> should be replaced once the cylinder pressure has dropped to 2 MPa (WMO report No. 292). Leuenberger et al. (2015) and Schibig et al. (2018) recommended that the usage of standard gas mixtures should be restricted to pressures above 3 MPa to remain within the WMO's compatibility goal of 0.1 μmol mol<sup>-1</sup> for the northern hemisphere and 0.05 μmol mol<sup>-1</sup> for the southern hemisphere. However, the CO<sub>2</sub> enrichment shown in Fig. 4b depends only on relative pressure, not absolute pressure, suggesting that determining the minimum operating pressure by considering the absolute pressure is not efficient. For example, if the initial pressure is low, the standard gas mixture will be replaced at a pressure at which it should have been usable, resulting in waste of the standard gas mixture. If the initial pressure is high, the standard gas mixture will not be replaced at the pressure at which it should be replaced, leading to poor consistency because of overestimation or underestimation of the observed values. Therefore, we recommend that the WMO's compatibility goal should be modified so that laboratories use the relative pressure as a criterion. If the CO<sub>2</sub> molar fraction is allowed to increase to 0.05 mol mol<sup>-1</sup>, the standard gas mixture should be replaced when the cylinder pressure drops to 3 MPa, 2 MPa,

- or 1 MPa for initial pressures of 15 MPa, 10 MPa, and 5 MPa, respectively. In this way, the standard gas
- mixture can be used efficiently without waste.

- The question arises as to whether the cylinder should be positioned horizontally or vertically during
- measurement. From Fig. 5, it appears to be best to operate a horizontally positioned cylinder with an
- outflowing gas flow rate of between 0.15 L min<sup>-1</sup> and 0.30 L min<sup>-1</sup>, because it may not be necessary to pay
- attention to deviations of CO<sub>2</sub> levels in the cylinders when taking out the standard gas mixture. However,
- a lower flow rate such as 0.080 L min<sup>-1</sup> may be desirable if the same set of standard gas mixtures is used
- for a long time. In that case, a correction would be necessary to ensure long-term consistency of the CO<sub>2</sub>
- molar fraction because the CO<sub>2</sub> deviation as the pressure drops cannot be ignored. Our results showed that
- the relative pressure determines the amount of CO<sub>2</sub> deviation, provided that the cylinder position and the
- gas outflow rate are constant. Therefore, CO<sub>2</sub> deviation could be corrected by determining in advance the
- relationship between CO<sub>2</sub> deviation and relative pressure and the flow rate of the outflowing gas. It should
- be noted, however, that this method is for correction of the CO<sub>2</sub> molar fraction in the cylinder, not correction
- of the positive and negative offsets in outflowing gases. The offset values should be corrected using a
- 15 fractionation factor as determined in Sect. 3.2.3. However, this correction may not be very useful, as the
- offsets would be at the same level as the measurement uncertainty. Indeed, it is important to note that the
- 17 atmospheric CO<sub>2</sub> molar fraction is difficult to determine with an uncertainty of less than 0.01 μmol mol<sup>-1</sup>
- due to the thermal diffusion effect.
- From the above discussion, the standard gas mixture should be operated during observation as follows.
- 1. The flow rate of outflowing gas from the cylinders should be as low as possible to reduce the
- contribution of thermal diffusion fractionation, although other effects, such as absorption/desorption
- in pressure regulators, should be also taken into account.
- 2. Throughout the observation, cylinders should be used in either a horizontal or a vertical position, and
- the position of the cylinders should not be altered.

 The cylinder pressure at which a standard gas mixture should be changed should be determined based on the relative pressure.

3

4

1

2

#### 5 Conclusions

5 We attempted to quantitatively estimate the factors that cause the CO<sub>2</sub> molar fraction in a cylinder to deviate 6 as the pressure drops, to facilitate a shift from the use of standard gas mixtures based on empirical 7 knowledge to use based on theoretical understanding. We found that the CO<sub>2</sub> molar fraction in the cylinder 8 changes from the initial value as a result of thermal diffusion fractionation as well as adsorption and 9 desorption. We found that thermal diffusion fractionation operates even at low gas outflow rates, for which 10 adsorption/desorption effects had been considered to be the main cause. A further important finding was 11 that this deviation of the CO<sub>2</sub> molar fraction is independent of the initial pressure and depends on the relative 12 pressure rather than the absolute pressure. Our results demonstrate the necessity for a new way of operating 13 cylinders that is different from the conventional empirical knowledge of the use of standard gases. 14 Furthermore, long-term consistency of values will be ensured by correcting for deviations in the CO<sub>2</sub> molar 15 fraction due to the pressure drop. Thus, this study is an important contribution to ensuring the consistency 16 of observed values, which has been a concern in long-term CO<sub>2</sub> observations.

- Data availability. The data presented in this article are available upon request to Nobuyuki Aoki (aoki-
- nobu@aist.go.jp).
- Author contributions. NA designed the study. NA performed the experiment and prepared the first draft.
- SI performed mass spectrometry measurements. Both authors contributed to the preparation of the final
- version of the manuscript.

# 6 Competing interests

The authors declare that they have no conflict of interest.

#### 8 Acknowledgments

- This study was partly supported by the Global Environment Research Account for the National Institutes
- of the Ministry of the Environment, Japan (grant nos. METI1454 and METI1953) and Japan Society for
- the Promotion of Science KAKENHI grants (grant numbers 19K05554 and 22H05006).

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
