# Peer review of "CO2 deviation in a cylinder due to consumption of a"

_EGUsphere, 2025_

## Author Response (AR1)

We appreciate your review and feedback on our manuscript. We acknowledge that their contributions will improve the quality of this study.

Our responses to the reviewers' comments are as follows:

**Response to RC1**

While IRMS analyses described within this work are sounds measurements the connection to thermal diffusion is unclear. Exclusion of this section still provides a quantifiable difference in amount fraction based on position of the cylinder and the flow rate.

Response: We appreciate your comment regarding the connection between the IRMS analyses and thermal diffusion. The decanting experiment allowed us to quantify deviations from CO2 amount fraction at low flow rates for vertically positioned cylinders; however, it was insufficient to conclusively determine the presence or absence of an offset at low flow rates. To address this limitation, we conducted IRMS analyses, which confirmed a positive offset at low flow rates in vertically oriented cylinders, as detailed in Section 3.2.3. As you correctly noted, the relationship between these IRMS results and thermal diffusion was not clearly described. In response, we have added a clarifying statement regarding this connection in the revised manuscript (page7, line 3–5, page 17, line 17–18, page 20, line 5–6 and lines 10–22).

If isotopic composition is to be explored, inclusion of delta 13C-CO2 and delta 18O-CO2 would be more constructive as it is directly applicable to measurements at hand. Additionally, these analyses were performed using an optical analyzer only quantifying 12C-CO2, it is possible that some of the difference seen could be based on release of one isotopologue over the other.

**Response**: Thank you for highlighting the potential influence of isotopic composition on the results. We acknowledge that the use of an optical analyzer that primarily detects  $^{12}C^{16}O_2$  could, in principle, lead to bias if significant isotopic fractionation due to the adsorption and thermal diffusion fractionation effects were present. However, the adsorption effect is governed by van der Waals forces. The intermolecular van der Waals forces are determined by the electronic structure of molecules—such as their polarity and polarizability. Their electronic structures between isotopologues are essentially identical. As a result, isotopic fractionation due to the adsorption is negligible. Furthermore, Thermal diffusion fractionation effect was roughly estimated based on  $\delta(^{29}N_2/^{28}N_2)$  value because deviation of  $\delta(^{13}CO_2)$  can be approximated from that of  $\delta(^{29}N_2/^{28}N_2)$ . A sentence addressing this point has been included in the revised manuscript (page 23, line 23 – page 24, line 18).

The blending process described on page 5, lines 7 - 12, sounds like the CO2 and air were housed in

separate containers and not mixed together. A gentle change in wording could make this clear for a broader audience.

**Response**: We revised the text to make the blending process clear (page 5, lines 11 – page 6, line 3)

Mole and molar fraction are both used within the text.

Response: We unified mole and molar fraction used within the text into molar fraction.

Page 6, line 9, the spelling for the instrument needs corrected from "Picaro" to "Picarro"

Response: The spelling of "Picaro" on page 6, lines 16 and 20 has been corrected to "Picarro."

Equations included after page 7 no longer include commas

**Response**: We added commas to the equations included after page 7.

Page 14, line 9, includes "quantitively", I recommend changes this to "quantitatively".

**Response**: We revised from "quantitively" to "quantitatively" according to your comment (page 13, line 16).

Figures 6 and 8 have different formatting from others.

**Response**: The formatting of figure 6 and 8 were revised.

**Response to RC2**

The manuscript "CO2 deviation in a cylinder due to consumption of a standard gas mixture" by Nobuyuki Aoki and Shigeyuki Ishidoya quantifies CO2 deviations in standard gas cylinders due to adsorption/desorption and thermal diffusion effects. The paper is well written and structured. Overall, it is sufficiently original and contains relevant, novel information to merit publication in AMT, provided some minor issues are addressed.

**General comments**

The measurement technique used only detects the main isotopes of CO2, which, as the other referee pointed out, influences the results presented in this work. However, this influence is probably small when considering amount fraction measurements of CO2. Nevertheless, I encourage the authors to mention this and, if possible, provide an estimate of the degree to which the results could be biased as a result.

Response: Thank you for highlighting the potential influence of isotopic composition on the results. We acknowledge that the use of an optical analyzer that primarily detects  $^{12}C^{16}O_2$  could, in principle, lead to bias if significant isotopic fractionation due to the adsorption and thermal diffusion fractionation effects were present. However, the adsorption effect is governed by van der Waals forces. The intermolecular van der Waals forces are determined by the electronic structure of molecules—such as their polarity and polarizability. Their electronic structures between isotopologues are essentially identical. As a result, isotopic fractionation due to the adsorption is negligible. Furthermore, thermal diffusion fractionation effect was roughly estimated based on  $\delta(^{29}N_2/^{28}N_2)$  value because deviation of  $\delta(^{13}CO_2)$  can be approximated from that of  $\delta(^{29}N_2/^{28}N_2)$ . A sentence addressing this point has been included in the revised manuscript (page 23, line 23 – page 24, line 18).

At the end of the paper (Page 27, lines 8-13), recommendations regarding the positioning, flow rates and the cylinder pressure are made. These recommendations are sound; however, I would exercise caution regarding the recommendation to use the lowest possible flow rates, as other effects, such as absorption/desorption in pressure regulators, may play an important role in practice. As these cannot be ruled out, they should be mentioned in the paper.

**Response**: We revised the sentence to "The flow rate of outflowing gas from the cylinders should be as low as possible to reduce the contribution of thermal diffusion fractionation, although other effects, such as absorption/desorption in pressure regulators, should be also considered" (page 28, line 21-22)

**Specific comments**

Page 5, lines 4-7: The calibration procedure is unclear.

**Response**: We revised the calibration procedure (Page 5, lines 5-11)

Page 11, lines 12-14: Cylinder orientation should be mentioned.

It is not clear to me whether the top, middle or bottom positions for temperature measurements were the same for horizontally and vertically positioned cylinders. It would be helpful to include a small illustration.

**Response**: Positions for temperature measurements were added in the Figure 1.

Figure 5: The quality of the figure is poor and needs to be improved.

**Response**: The quality of figure 5 was improved.